# Covid19: Unless one gets everyone to act, policies may be ineffective or even backfire

**Alessio Muscillo**[1], **Paolo Pin**[1,2]*, **Tiziano Razzolini**[1,3]

**1** Dept. Economics and Statistics, University of Siena, Siena, Italy, **2** IGIER & BIDSA, Bocconi University, Milan, Italy, **3** IZA, Bonn, Germany

* paolo.pin@unisi.it

**Data Availability Statement:** All relevant data are within the paper and its Supporting Information files.

## Abstract

The diffusion of Covid-19 has called governments and public health authorities to interventions aiming at limiting new infections and containing the expected number of critical cases and deaths. Most of these measures rely on the compliance of people, who are asked to reduce their social contacts to a minimum. In this note we argue that individuals' adherence to prescriptions and reduction of social activity may not be efficacious if not implemented robustly on all social groups, especially on those characterized by intense mixing patterns. Actually, it is possible that, if those who have many contacts have reduced them proportionally less than those who have few, then the effect of a policy could have backfired: the disease has taken more time to die out, up to the point that it has become endemic. In a nutshell, unless one gets everyone to act, and specifically those who have more contacts, a policy may even be counterproductive.

## Introduction

As social scientists, we use epidemic models to mimic the diffusion of opportunities and ideas in the society. In this context, we are used to think of the effects of people's choices and actions on diffusion processes, like viral marketing campaigns or the launch of new technologies that increase online contacts. In general, our focus is on what happens when each individual in the society takes autonomous decisions that affect her socialization. Looking at people's responses and decisions can be helpful to design policies against diseases and to understand how they affect the behavior of other members of the society.

Following the outbreak of the new Coronavirus, governments have faced the necessity to foster the limitation of social contacts. In most cases, initially the population have been asked to limit their contacts relying on the individual sense of responsibility (an extreme case was the initial approach in the United Kingdom, where isolation was intended only for people suspected to be infected or arrived from abroad, as stated by the Health Protection (Coronavirus) Regulations 2020 on March 10); while only at a later stage rigid temporary laws have been issued (like China on January 23 and 26, 2020, or Italy on March 4 and 11). However, independently of the nature of the restrictions, it is clear that not every individual responds in the same way to impositions and requests. Some people cut immediately all their social contacts, while

**Funding:** We gratefully acknowledge funding from the Italian Ministry of Education Progetti di Rilevante Interesse Nazionale (PRIN) grant 2017ELHNNJ.

**Competing interests:** The authors have declared that no competing interests exist.

others may only marginally reduce them. The classic argument of *revealed preferences* suggests that those who have more social relations will be less prone to limit them: their behavior reveals that they care more than others about social interactions, for personal taste or for professional reasons. So, if we just ask people to reduce their contacts at a level they feel safe, everybody will trade off the expected risks with the benefit that they perceive from socialization. Thus, those who have many contacts every day will be proportionally less inclined to cut them, compared than those who have few.

Available data and public concern often focus on the number of contacts that people have, with the obvious implication that a reduction in the average number of contacts would correspond to a reduction in the infections. In this note, we use an empirical analysis and a stylized model to show that, together with the average number of contacts, other measures of statistical dispersion—i.e. squared number of contacts (roughly, variance)—are also important to explain the variation in the number of infections.

## Empirical motivation

We conduct an exploratory analysis using the public dataset provided by Belot and colleagues [1]. This dataset contains, among other information, survey data on the individual number of contacts in the regions of six countries before and after the interview, occurred in the third week of April 2020. The respondents to the survey were asked the number of their contacts before the outbreak of Covid-19 and in the last two weeks preceding the interview. We have then computed the average number of contacts and average squared number of contacts at the regional level.

We limit our analysis to the regions of Italy, South Korea and the United Kingdom, because of the large impact of the coronavirus in these countries and of the availability of data at a regional level. For each region, we have collected the weekly number of cases and deaths before and after the survey interview using the publicly available datasets listed in the S1 Document.

Italy and South Korea displayed a similar timing in the diffusion of coronavirus as well as in government interventions, as shown by the *government response stringency index* developed by Hale and colleagues [2]. For these two countries we have computed the weekly number of confirmed cases and deaths in the week from March 1 to March 8 for the pre-interview period and in the week from April 13 to April 20 for the post-interview period. Due to the later diffusion of coronavirus in the UK, the weekly number of confirmed cases and deaths is measured during the week from March 13 to March 20 and the week from April 24 to May 1 (in the UK, the indexes on the diffusion of Covid-19 are reported on a weekly basis, starting on Fridays).

We perform two ordinary least squares regressions where the dependent variables are the regional pre- and post-interview variation in the number of confirmed cases and deaths (respectively, ΔConfirmed Cases and ΔDeaths). The number of contacts that an individual has is denoted by $d$, so that the (regional) average number of contacts is denoted by $\langle d \rangle$ and the (regional) average squared number of contact is $\langle d^2 \rangle$. The explanatory variables used in the regression are the pre- and post-interview variation: $\Delta\langle d \rangle$ and $\Delta\langle d^2 \rangle$.

The estimates are shown in Table 1. Columns (1) and (2) report the estimated effects on the variation in the number of confirmed cases while columns (3) and (4) the estimated effects on the variation in the number of deaths. The variation in the number of contacts, $\Delta\langle d \rangle$, always displays a negative effect significantly different from zero on both dependent variables, as an obvious result of the endogenous reaction to the spread of coronavirus. This is because causality in the real data goes in both directions, and the larger is the diffusion of the virus, the larger is the average reduction in contacts. The variation in the squared number of contacts, $\Delta\langle d^2 \rangle$, has a positive effect always statistically different from zero at 5% significance level. The inclusion of the latter regressor always improves the adjusted $R^2$, thus indicating that this variables has a

**Table 1. Variations in confirmed cases and deaths.**

| | Δ Confirmed Cases | | Δ Deaths | |
|---|---|---|---|---|
| | **(1)** | **(2)** | **(3)** | **(4)** |
| $\Delta\langle d\rangle$ | -69.228*** | -167.411*** | -25.549*** | -50.594*** |
| | (21.289) | (51.019) | (8.300) | (14.582) |
| $\Delta\langle d^2\rangle$ | | 0.382** | | 0.097** |
| | | (0.172) | | (0.044) |
| Constant | 213.419** | 163.936 | 92.514*** | 79.892** |
| | (102.497) | (113.241) | (33.942) | (33.278) |
| N. obs. | 48 | 48 | 48 | 48 |
| Adj. $R^2$ | 0.100 | 0.231 | 0.158 | 0.245 |

In columns 1 and 2 the dependent variable is the variation in the numbers of confirmed cases in a region. In columns 3 and 4 the dependent variable is the variation in the number of deaths in each region. $\Delta\langle d\rangle$ is the variation in the average number of contacts in each region. $\Delta\langle d^2\rangle$ is the variation in the average squared number of contacts in each region. Robust standard errors in parentheses.

*** significant at 1%,

** significant at 5%,

* significant at 10%.

great explanatory power. In the S1 Document we also test the presence of non-linear effects including as a regressor the variation in the square of the average number of contacts.

## SI–type model

In this section, we build a stylized model of epidemics on networks [3, 4] and, in contrast to the previous literature, consider the possible negative effects of a quarantine that is not homogeneous [5–7]. We argue that when a disease spreads in a population with heterogeneous intensity of meetings—a so-called *complex network*—if the individuals who meet many people exhibit high resistance against isolation policies, such policies may not only turn out to be ineffective, but can even be detrimental. Imagine a social-distancing policy that asks people to limit their contacts to reduce the diffusion of a disease. This generates a new reduced social network that is smaller but denser. The main unintended negative consequence of the policy could be that even if the disease was eventually going to die out in the original social network, it becomes endemic in the new network instead. Paradoxically, as long as the policy is in force, the disease will be kept alive.

The intuition behind the phenomenon is simple to grasp, and we leave the details of a parsimonious susceptible-infected-susceptible (SIS) model in the S1 Document. Consider the social network of a society, where some people have few links and others have many. Consider, also, a disease spreading via these contacts. Whether the disease will be *endemic* or not turns out to depend on the interplay between the features of the disease itself and the statistical properties of the social network through which it is spreading. The disease is concisely described by the transmission rate, $\beta$, and the recovery rate, $\delta$. The characteristics of the social network are captured by the number of contacts that one has $d$. In particular, by its average across people, denoted $\langle d\rangle$, and by the expected square of this number, $\langle d^2\rangle$.

Whether the disease dies out or remains endemic depends on the relationship between two quantities:

$$\lambda = \frac{\beta}{\delta} \quad \text{and} \quad \mu = \frac{\langle d\rangle}{\langle d^2\rangle}. \tag{1}$$

A high $\lambda$ indicates a disease that is highly contagious and slow to recover from. On the contrary, $\mu$ describes the heterogeneity of the network. The analysis of the model shows that $\mu$ captures how much the structure of the network slows diffusion processes down: the lower the $\mu$, the more dangerous the situation is (with a physics analogy, $1/\mu$ can be though of as the *conductivity* of the network with respect to the disease's diffusion process). When $\lambda < \mu$ the disease is not endemic, and the difference between them tells us how fast it will die out. Instead, when $\lambda \geq \mu$, the disease becomes endemic.

Social distancing policies aim at reducing the contacts among people, thus modifying the original social network in order to cut or interrupt the transmission chain of the disease, until it dies out. However, if not everyone responds in the same way to the policy, then the resulting network may turn out to be sparser on average, but still too dense of contacts among the most active individuals. This can happen, for instance, if those who have more contacts are relatively less responsive to the policy indications. Unfortunately, then, the new smaller network might have some properties, such as a low $\mu$, that might hinder the containment of the disease or even "help" the disease to remain endemic among those individuals who keep on being active.

Imagine, for example, that the number of people that one meets on a daily basis ranges from five to 50. As usually happens in the real world (see Fig 1), many people have few connections while few individuals, called *hubs*, have a lot more. Now, say that everybody is asked to cut their meetings by the same quantity (to begin with, just three contacts, which is more than half for peripheral nodes, but proportionally very little for the hubs), so that the new *degree distribution* is shifted down, but keeps the same variance. In this case, a $\mu$ that was originally higher than $\lambda$ may actually decrease, so that a disease may remain active for more time. If more contacts are dropped with the same uniform rule, $\mu$ may keep decreasing, up to the point that it becomes smaller than $\lambda$, and the disease remains endemic in the society, at least as long as the population is in the new network exhibits $\mu < \lambda$. This remains true even if an additional

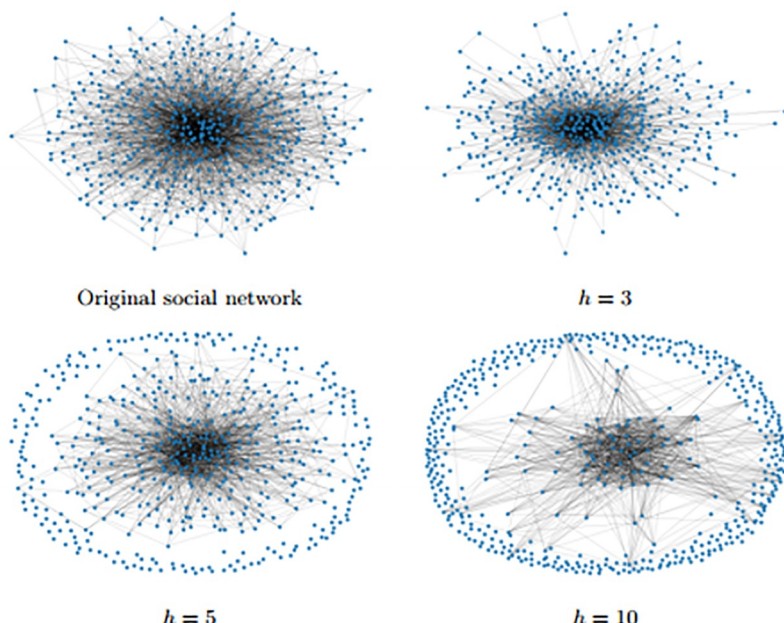

**Fig 1. A social network of 500 nodes.** This is a social network with degree distribution given by P(5) = P(10) = 0.4, P(20) = 0.1, P(40) = P(50) = 0.05. Different self-isolation measures are applied, depending on the number *h* of links removed to each node. In this example $\mu$ falls down as *h* increases.

cut completely isolates some nodes in the network, or even most of them. The sub–population of the remaining ones, those who had originally many links and are still very connected, will behave as an incubator for the disease because they form now a denser sub–network.

By contrast, a policy that imposes a proportional cut their contacts to each individual always delivers an increased $\mu$ (e.g. it doubles if the reduction is by 50% for all nodes).

## Conclusion

In this note, we argue that a valid social-distancing intervention by the authorities should play on both the uniform scaling that reduces contacts by a constant amount (as is the effect of closing schools for students) and on targeting those individuals with many contacts (which could be obtained by closing or regulating private activities like shops and leisure meeting points).

Our claim is based on a simple SIS model (see S1 Document) and consistent with the empirical analysis shown in Table 1. It is also in line with other models that have been recently proposed, as in the examples from the Stanford Human Evolutionary Ecology and Health group or the SIR models by Anderson and colleagues [8] and Koo and colleagues [9]. The same message comes from the empirical work of Chinazzi and colleagues [10], analyzing human mobility data from airline companies. All these works point out that restrictions are effective only if everyone fulfills the prescriptions and limits socialization.

Several issues are not dealt with here, such as mental health consequences of the imposed isolation [11–13] and the impact on the capacity of the health systems, because the specific focus of this work is on proposing more efficient social-distancing policies. To do so, we highlight the importance of distinguishing people by their degree of socialization, and remark that if not everybody reduces drastically and proportionally their social contacts, then such measures could have an effect opposite to the one expected.

## Supporting information

**S1 Document. Empirical analysis, data on number of contacts, S1-S3 Tables, data on number of cases/deaths, robustness check, and the model.**
(PDF)

**S1 File. Stata code for data analysis.**
(DO)

**S2 File. Python code for figures.**
(PY)

## Acknowledgments

We thank Alberto Dalmazzo, Matthew Jackson and Alessia Melegaro for very helpful comments. We gratefully acknowledge funding from the Italian Ministry of Education Progetti di Rilevante Interesse Nazionale (PRIN) grant 2017ELHNNJ.

## Author Contributions

**Conceptualization:** Paolo Pin.

**Data curation:** Tiziano Razzolini.

**Formal analysis:** Alessio Muscillo, Paolo Pin, Tiziano Razzolini.

**Investigation:** Paolo Pin.

**Methodology:** Alessio Muscillo, Paolo Pin.

**Supervision:** Paolo Pin.

**Visualization:** Alessio Muscillo, Paolo Pin.

**Writing – original draft:** Alessio Muscillo, Paolo Pin.

**Writing – review & editing:** Alessio Muscillo, Paolo Pin, Tiziano Razzolini.

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
