## [Decision Letter · Decision Letter 0]

21 May 2020

PONE-D-20-09143

Covid19:

unless one gets everyone to act, policies may be ineffective or even backfire

PLOS ONE

Dear Professor Pin,

Thank you for submitting your manuscript to PLOS ONE. After careful consideration, we feel that it has merit but does not fully meet PLOS ONE’s publication criteria as it currently stands. Therefore, we invite you to submit a revised version of the manuscript that addresses the points raised during the review process.

We would appreciate receiving your revised manuscript by Jul 05 2020 11:59PM. To enhance the reproducibility of your results, we recommend that if applicable you deposit your laboratory protocols in protocols.io, where a protocol can be assigned its own identifier (DOI) such that it can be cited independently in the future. For instructions see: http://journals.plos.org/plosone/s/submission-guidelines#loc-laboratory-protocols

We look forward to receiving your revised manuscript.

Kind regards,

Angelo Brandelli Costa

Academic Editor

PLOS ONE

Journal Requirements:

Additional Editor Comments (if provided):

Reviewers' comments:

Reviewer's Responses to Questions

**Comments to the Author**

1. Is the manuscript technically sound, and do the data support the conclusions?

Reviewer #1: Yes

Reviewer #2: Yes

Reviewer #3: Partly

2. Has the statistical analysis been performed appropriately and rigorously? 

Reviewer #1: N/A

Reviewer #2: Yes

Reviewer #3: Yes

3. Have the authors made all data underlying the findings in their manuscript fully available?

Reviewer #1: Yes

Reviewer #2: Yes

Reviewer #3: Yes

4. Is the manuscript presented in an intelligible fashion and written in standard English?

Reviewer #1: Yes

Reviewer #2: Yes

Reviewer #3: Yes

5. Review Comments to the Author

Reviewer #1: In this paper the authors discuss the impact of COVID-19 spreading on social groups, focusing on those who are heterogeneous in terms of social interactions (people with few and people who have more social relations and interactions). To this end, they proposed a suscetible-infected-suscetible SIS model, considering features such as transmission and recovery rates of a disease and the contact numbes of individuals.

As conclusion, the authors argue that when a disease spreads in a population with heterogeneous levels of social meetings, if the individuals who meet many people dont fully respect isolation policies, such policies can be ineffective or damaging.

The study is interesting and could help governments and health authorities to propose more efficient social isolation policies. I agree that if not everyone does their part by maintaining social isolation as long as possible, the desease may remain active for more time. However, when considering the pandemic caused by Corona virus, even if not all people reduce social interactions to the minimum possible, that still may contributes to reducing the number of patients simultaneously infected, preventing collapses in health systems.

While the study appears to be sound, I miss some validation of the model. Would it be possible to include in the article some data related to the spread of COVID-19 observed in some countries, such as the contagion rate, in contrast to the different rates of social isolation in each of them? That would be nice and could corroborate the conlusion of the study.

The authors mentioned some related work (references 6-8). I suggest including a brief description of the related work. Even though the authors claim that the proposed model is different and consider other features, that description could

I also suggest, if possible, to compare the proposed model with the others, allowing to assess the impact of heterogeneous quarantine, once that feature was not considered in the related.

Reviewer #2: "This article is on a feature that must be highly considered in the context of pandemic COVID-19/SARS-COV2, which regards social/physical distancing/isolation to slow the spread of the infection, avoid the colapse of health systems and buy some time for researchers come up with specific, effective treatment and/or vaccine. Since there is evidence toward distancing/isolation to slow the spread of a highly contagious disease, it seems this article boosts the recommendation of such measure and provides endorsing mathematical and simulation models. There is ongoing discussion, though, on the mental health consequences of distancing/isolation which is inevitable and should also be considered, and that health care systems/policies and providers should prepare to minimize its impact on the population, as discussed in JAMA (doi 10.1001/jamapsychiatry.2020.1057 and 10.1001/jamainternmed.2020.1562) and NEJM (doi 10.1056/NEJMp2008017)."

Reviewer #3: The manuscript provides an innovative approach using a susceptible-infected-susceptible (SIS) model to advocate a "proportional reduction" on social contacts in order to contain new SARS-Cov-2 new infections on populations. The authors proposed a parsimonious SIS models to sustain that when a disease spreads in a given population with heterogeneous intensity of meetings, policies advocating homogeneous social contacts may be ineffective and even deleterious.

The manuscript contains some colloquial expressions and statements such as: "In a nutshell, unless one gets everyone to act, and specifically those who have more contacts, a policy may even be counterproductive". The conclusions of the paper seems a little vague: "The specific focus of our approach is to distinguish people by their degree of socialization, and remark that if not everybody reduces drastically and proportionally their social contacts, then such measures could have an effect opposite to the one expected."

The authors state that previous published works have focused on models in a context of quarantine bud did not consider the "possible negative effects" caused by the heterogeneity of the quarantine, bud provide few empirical data to sustain such a statement.

There's no limitation section on the manuscript describing the potential limitations and pitfalls of the provided model. The appendix provide the mathematical notation of the model and its applications but again, providing no empirical data. Although one can argue in defense of such a statistical note, it is worth considering that this approach may not be exactly of interest to the Plos One's audience.

6. PLOS authors have the option to publish the peer review history of their article (what does this mean?). If published, this will include your full peer review and any attached files.

Reviewer #1: No

Reviewer #2: No

Reviewer #3: No

---

## [Author Response · Author response to Decision Letter 0]

17 Jun 2020

We have added a pdf with detailed responses to the three reviewers.

---

## [Decision Letter · Decision Letter 1]

21 Jul 2020

Covid19:

unless one gets everyone to act, policies may be ineffective or even backfire

PONE-D-20-09143R1

Dear Dr. Pin,

We’re pleased to inform you that your manuscript has been judged scientifically suitable for publication and will be formally accepted for publication once it meets all outstanding technical requirements.

Kind regards,

Angelo Brandelli Costa

Academic Editor

PLOS ONE

Additional Editor Comments (optional):

Reviewers' comments:

Reviewer's Responses to Questions

**Comments to the Author**

1. If the authors have adequately addressed your comments raised in a previous round of review and you feel that this manuscript is now acceptable for publication, you may indicate that here to bypass the “Comments to the Author” section, enter your conflict of interest statement in the “Confidential to Editor” section, and submit your "Accept" recommendation.

Reviewer #1: All comments have been addressed

Reviewer #3: All comments have been addressed

2. Is the manuscript technically sound, and do the data support the conclusions?

Reviewer #1: Yes

Reviewer #3: Yes

3. Has the statistical analysis been performed appropriately and rigorously? 

Reviewer #1: Yes

Reviewer #3: Yes

4. Have the authors made all data underlying the findings in their manuscript fully available?

Reviewer #1: Yes

Reviewer #3: Yes

5. Is the manuscript presented in an intelligible fashion and written in standard English?

Reviewer #1: Yes

Reviewer #3: Yes

6. Review Comments to the Author

Reviewer #1: The suggested changes were duly addressed in the new version of the paper. The improvements to the text have made it more understandable.

Reviewer #3: The points identified by me during the first review were answered by the authors. The Empirical Motivation's session got a substantial admendment.

7. PLOS authors have the option to publish the peer review history of their article (what does this mean?). If published, this will include your full peer review and any attached files.

Reviewer #1: No

Reviewer #3: No

---

## [Editor Report · Acceptance letter]

24 Aug 2020

PONE-D-20-09143R1 

Covid19:
unless one gets everyone to act, policies may be ineffective or even backfire 

Dear Dr. Pin:

I'm pleased to inform you that your manuscript has been deemed suitable for publication in PLOS ONE. Congratulations! Your manuscript is now with our production department. 

Kind regards, 

on behalf of

Dr. Angelo Brandelli Costa 

Academic Editor

PLOS ONE